# Choosing to succeed? Insights into doctoral students' supervisor selection and its outcomes

Danila Pavliuk *, Svetlana Zhuchkova

Center for Sociology of Higher Education, Institute of Education, HSE University, Moscow, Russia

* dmpavlyuk@hse.ru

## Abstract

Doctoral student experience and outcomes largely depend on the quality of supervision. Research on doctoral supervision usually focuses on the attributes, styles, and roles of a supervisor during doctoral training, while the process and effects of initiating supervisor-supervisee relationships remain less explored. This paper aims to examine how students' strategies for supervisor selection are related to subsequent difficulties in interaction with their supervisors and their confidence about their future dissertation defense. Based on a nationwide survey of doctoral students in Russia (N = 1796) and regression analysis, we demonstrate that academic criteria such as a supervisor's subject-matter expertise and successful supervision history are the best factors for students to consider to reduce the probability of future difficulties with the supervisor and to avoid experiencing a lack of confidence. Using an existing positive personal relationship with a supervisor as a criterion reduces the chances of interaction difficulties, but does not affect a student's confidence in future academic success. At the same time, the "only option" scenario (i.e., a situation when students have no alternatives to choose from) is associated with higher chances of facing difficulties during the doctoral journey and a lower level of student confidence. The study highlights the critical role of agency in supervisor selection, showing that structured support for informed decision-making – whether through institutional matchmaking systems or student mentoring programs – can mitigate risks of poor supervision outcomes. These findings call for policy interventions to promote transparency in supervisor selection processes, especially in contexts where students have limited choices. The results of the research can be used by prospective doctoral students and doctoral program administrators to create a personal or institutional navigation system for choosing a supervisor.

## Introduction

"Supervision is the most variable of all variables" [1], as Bowen and Rudenstine wrote in the early 90s. Indeed, supervision is consistently considered a key component in

**Data availability statement:** We have published regression dataset that used in our research at Harvard Dataverse under CC0 v1. Pavliuk, Danila; Zhuchkova, Svetlana, 2024, "Replication Data for research paper: Choosing to succeed? Insights into doctoral students' supervisor selection and its outcomes", https://

doi.org/10.7910/DVN/YKNXO3, Harvard Dataverse, V1

**Funding:** This study was financially supported by the Basic Research Program at the National Research University Higher School of Economics (HSE University) in the form of research-grant funding received by DP and SZ. No additional external funding was received for this study. The funder had no role in study design, data collection and analysis, decision to publish, or preparation of the manuscript.

**Competing interests:** The authors have declared that no competing interests exist.

the success of a doctoral student [2–4]. Research demonstrates that positive supervision experiences increase satisfaction, enhance research progress, and maintain the psychological well-being of doctoral students [5,6]. Conversely, imbalanced and inadequate academic advising may have the opposite effect, degrading students' experiences [7]. Such varied outcomes have led national education systems and the academic community to become keenly interested in enhancing the quality of supervision [8–10].

Previous research in this area has focused on the styles, functions, and roles of academic supervisors in a doctoral journey [4,11,12]. These studies have primarily examined student-supervisor interactions during doctoral training, but there is scant literature on prior relationships between students and supervisors, including how supervisors are chosen [13]. A poor fit can lead to misalignment of expectations in supervisee-supervisor relationships, affecting further doctoral experiences and outcomes [14].

Few studies have focused on the factors influencing this choice. The choice may be based on rational calculations or the expected behaviors of the supervisor, but it may also rest on irrational foundations [12]. Researchers typically emphasize a supervisor's academic attributes, reputation, overall supervision experience, and demographic factors, including age, gender, race, and nationality, as the most frequently used criteria for selecting a supervisor [6,12,15]. However, most studies have identified factors affecting selection but have not assessed the relationship between these factors and students' satisfaction with their supervisory interactions or their academic results. Understanding this linkage is crucial because early-stage selection decisions may either facilitate smooth progress or lead to irreversible conflicts, ultimately impacting completion rates and research quality. Our study addresses this gap by examining whether various selection strategies contribute to difficulties in interaction with a supervisor during doctoral training and students' confidence that they will successfully defend a dissertation. Using data from a nationwide survey of doctoral students in Russia conducted in 2022, we aim to answer the following research questions:

*RQ1: Are different supervisor selection strategies related to students' subsequent difficulties in interaction with a supervisor?*

*RQ2: Are different supervisor selection strategies related to students' confidence in a successful future defense?*

## Navigating selection strategies: A literature review

Relationships between doctoral students and their supervisors play a crucial role in the quality of the doctoral experience, the completion of a dissertation, and the overall well-being of doctoral students [4,5,16]. Research often focuses on processes during doctoral training, but less attention is paid to the initial stages of student-supervisor relationship formation and development. However, studies have shown that compatibility between doctoral students and their supervisors, including academic interests and working styles, is significantly related to students' satisfaction and success [17,18]. Although in some countries and universities supervisors are

appointed by administrators, researchers note that the selection process is often informal, allowing doctoral students to influence it [18–20]. Thus, the "right" choice of supervisor is crucial for matching the expectations of both student and supervisor regarding work styles and necessary support [21]. Research shows that students involved in selecting their supervisors are more satisfied with their doctoral experience and demonstrate greater progress throughout their doctoral journey [18].

In this part, we review the criteria for choosing a supervisor highlighted in previous research, starting with the most common ones, related to academic attributes, and finishing with less frequently examined criteria, such as demographic characteristics and pragmatic benefits.

One of the most explored criteria in the selection of a supervisor is her or his academic attributes and expertise [13]. The importance of a supervisor's academic achievements and qualifications is particularly relevant to the specific topic of the dissertation, rather than in a general sense [15,22–24]. Research has shown that choices based on such criteria lead to student satisfaction when working with a supervisor [18], especially if supervisors are highly experienced in a specific research topic [6]. Supervisors with subject-matter expertise and general research experience can provide targeted guidance, anticipate methodological pitfalls, and offer relevant networks – all of which streamline the research process. Doctoral students usually evaluate a supervisor's academic experience based on their publication record, the number and size of their research grants, and their educational background [25,26].

Relevant research has demonstrated that doctoral students are also guided by previous interactions with their supervisors [12,27]. Established relationships built earlier become an important factor in the selection process. Typically, a positive relationship with a supervisor can enhance student experience and improve research and educational results [28,29]. In particular, this can be explained in a way that supervisor and student are already used to each other's working styles or they can already have a head start on the PhD thesis [30]. However, qualitative studies have investigated that previous positive interactions can have a negative effect if a supervisor fails to provide adequate academic support [18,27]. Some studies have examined the negative consequences associated with mixing academic and non-academic interests, such as romantic relationships between supervisees and supervisors [31]. Such blurred boundaries can lead to conflicts of interest, compromised academic judgment, and power imbalances that can undermine the integrity of the supervision process [32].

A supervisor's previous supervision experience is also a significant choice factor [25]. Studies have highlighted the value of a supervisor's background in supervision as a positive factor for defending a dissertation [18]. This connection may be explained by the fact that experienced supervisors have a clearer understanding of the doctoral research process and requirements and a proven ability to guide students through complex academic and bureaucratic procedures [18,25]. Students may evaluate the supervision experience of faculty based on recommendations and reputation among previous students and colleagues within a department [15,33].

Some studies have investigated the significance of supervisors' demographic characteristics such as age, gender, race, and nationality when a student chooses a doctoral supervisor. Although some research shows these factors are meaningful when selecting supervisors at the bachelor's or master's levels [34], their importance appears to be comparatively lower for doctoral students, who tend to focus more on the substantive criteria described above [13]. Less frequently mentioned criteria relate to pragmatic benefits such as the supervisor's time availability, research funding, financial status, and administrative position at the university [25,34,35].

Although there is a plethora of research dedicated to selection criteria and strategies, few empirical quantitative studies investigate their consequences and effects. The studies reviewed in this section often face methodological limitations, such as being restricted to a single university or having a limited number of control variables, preventing broader conclusions about their relevance in diverse institutional contexts. Our study aims to advance the literature by analyzing data from a nationwide survey of doctoral students in Russia, offering a broader perspective on the selection criteria for doctoral supervisors and its effects. While our sample may not be fully representative, it enables us to explore a wider range

of factors related to supervisor selection simultaneously, taking into account other possible predictors of student success (which serve as control variables in our models). Understanding these relationships can provide valuable insights for policymakers and academic institutions in developing guidelines that could improve the impact of the supervisor selection process for doctoral students.

## Doctoral supervision in the context of Russian doctoral education

Doctoral education in Russia is provided by several types of organizations: universities (87% of the doctoral student body), research institutes (12%), and other organizations (less than 1%). According to the Federal State Statistics Service (2024), 45,075 students were enrolled in doctoral programs in 2022, while only 1,791 graduates were awarded a doctoral degree that year [36]. At the end of the 2021 academic year (a point close to the period when our data were collected), more than half (56%) of doctoral students in Russia across all types of organizations were male, and 59% held state-funded positions [37].

Doctoral education in Russia has been experiencing challenges and transformations during the past 15 years [38]. Alarming tendencies are related to the declining number of students (from 157,437 students in 2010–121,555 in 2023); the decreased number of graduates (from 28,273 graduates in 2014–14,146 in 2023); the increasing attrition rate (from 36% in 2014 to 48% in 2021 according to available data [39]); and the decreasing defense rate (from 28.5% of graduates defending their dissertations in the normative period of time in 2010 to only 11.2% doing so in 2023). The current state is described in the academic literature as the "crisis of Russian doctoral education" [40]. Although the reasons for this crisis are complex, poor supervision is considered one of the key contributing factors [38,40]. Despite the reform of transition from the traditional master-apprentice model to the structured model of doctoral programs, which took place in 2012–2013, supervisors are still the main figures in doctoral journeys in Russia. That is, they still provide all the guidelines on how to prepare and defend dissertations (as opposed to structured models where some of the necessary knowledge could be provided by school administrators or other faculty), which amplifies students' excessive dependence on them [41]. At the same time, various empirical studies show that approximately 16% of supervisors in Russia can be described as "hands-off supervisors." That is, they do not perform any functions associated with the supervisory role [42]. Practices of team supervision (i.e., cases when a student has several official supervisors) are rare [43] and are only possible in circumstances of multidisciplinary research or in programs arranged by two institutions [44]. Enhancement of supervision quality is the least frequent measure mentioned in the development programs of leading Russian universities [45]. Academics attribute the poor quality of supervision to a combination of factors, including supervisors' time limitations and high workload for other activities such as research and teaching, a lack of professional development programs, and insufficient incentives [46]. These circumstances are only worsened by high academic requirements imposed on doctoral students (for most fields of study, he or she should have at least three papers in highly ranked journals to get to a dissertation defense), as well as a high level of bureaucracy associated with the final stages of doctoral training, when the help of the supervisor is also necessary. Thus, defenses in Russia follow strict formal requirements mandated by the Higher Attestation Commission, including mandatory pre-defense procedures (which require departmental reviews, approval timelines, and formal website announcements), document verification at multiple administrative levels, standardized formatting rules for dissertation manuscripts, and so on. Usually, supervisors navigate these bureaucratic flows (i.e., interpret these requirements for candidates, coordinate with dissertation councils, and secure necessary approvals). Some universities have the right to award their degrees and can simplify these procedures, but their requirements regarding students' publications cannot be lower than the national ones.

Criteria and procedures for supervisor appointment in Russia are regulated at the national, institutional, and departmental levels [46]. National-level regulations provide minimal requirements for the qualifications and research productivity of a supervisor: he or she has to have a research degree and has to have conducted research on the thesis topic of a doctoral student in recent years, as well as have relevant publications in the previous three years. Additionally, a deadline is set for an official supervisor appointment, which is now 30 days after a student's enrollment (i.e., each student must

choose and secure approval for a supervisor within one month of starting the program). Institutional- and departmental-level regulations complement the national rules. To work as a supervisor, a candidate who meets the criteria should be approved by the corresponding department at the university. Doctoral students in Russia have the right to choose a supervisor, or one may be suggested by the department. While it is hard to empirically estimate the prevalence and diversity of practices at the stage of supervisor selection, research shows that only 19% of Russian universities require doctoral candidates to have preliminary agreements with their potential supervisor at the admission stage [47], indicating that the appointment process usually takes place after the student has enrolled. However, according to another empirical study based on survey data from Russian doctoral students, only 16% select their supervisors based on departmental recommendations, while the rest typically choose them independently [48]. This suggests that, in Russia, it is more common for students to choose their supervisors rather than having them assigned.

## Methods

### Sample

In this study, we used data from a nationwide survey carried out under the framework of the "Monitoring of Education Markets and Organizations" (MEMO) project. The project was supported by the Russian Ministry of Science and Higher Education (RMSHE). Data were collected from May 31 to August 31, 2022. The target population for the survey was students who studied in doctoral programs in Russian universities and research institutes in the 2021–2022 academic year.

Universities and research institutes were invited by the RMSHE to participate in the study and were requested to distribute a link to the questionnaire among their doctoral students. The survey was arranged online (Computer-Assisted Web Interviewing). Participation was voluntary. The data were obtained in an anonymized form to ensure that participants could not be identified during or after data collection. Informed consent was obtained at the beginning of the online survey. On the first page of the survey, respondents were required to check a box indicating their agreement to participate. This empirical study was reviewed and approved by the Institutional Review Board of the National Research University Higher School of Economics (HSE IRB). Before launch, the survey was piloted with a sample of 30 doctoral students from HSE University to check clarity, avoid ambiguous wording, and assess preliminary response patterns. Feedback from the pilot phase led to minor revisions.

The survey comprised 11 thematic blocks addressing the period before and during the doctoral studies, as well as expectations about the future. It also collected socio-demographic data.

Participants comprised a quota-based sample of 2392 doctoral students. To ensure that the sample was representative of various organizations and demographics, quotas were established based on organization type, federal district, university status, doctoral students' form of study and field of study. The attainment of quotas was monitored throughout the data-collection period, reminders were sent to underrepresented groups, and each organization was verified to have met its target quotas. The quotas did not take into account demographic characteristics such as age or gender. Consequently, caution should be exercised when generalizing the study's findings. Nevertheless, this survey is currently the only national dataset encompassing diverse categories of doctoral students across a variety of organizations and regions.

For our study, we focused on doctoral students enrolled at universities, excluding those from research institutes, who represented only 15% of our sample and 10–13% of all doctoral students in Russia. In Russia, the environment in research institutes differs significantly in comparison with universities, including less-formal relationships between supervisors and doctoral students, a higher level of inbreeding among doctoral students, lower variety of career trajectories, less-structured doctoral programs, and more active integration of students into real research projects [49,50]. Their specific characteristics and experiences were not the primary focus of our investigation. Additionally, fifth-year students, as well as students specializing in medicine and agriculture, were excluded from our analysis due to their small representation in the sample.

Ultimately, 1796 participants were included in our analysis. Among them, 40% studied at leading universities, i.e., universities that participate in the academic excellence program "Priority 2030" or hold at least one of the following special statuses: national research university, federal university, or member of the Association of the Leading Universities. Among other things, such statuses indicate that universities take steps to enhance the visibility, attractiveness, and performance of their doctoral programs. Over half (53%) of the sample were female students. Nearly two-thirds of the participants were first- (36%) and second-year (33%) students (with supervisors already assigned to them according to the national regulations). The majority (83%) were full-time students, and 71% had a tuition-free form of financing. In terms of the major fields of study, 25% were from the social sciences, 23% from technology, 23% from natural sciences and mathematics, 17% from the humanities, and 12% from educational sciences.

### Measures

**Supervisor selection strategies.** To identify various supervisor selection strategies, we used the following multiple-choice question (the percentage of participants who selected each option is presented in parentheses, N = 1796):

*Which of the following did you use as criteria when choosing a supervisor?*

1. *The authority and expertise of the supervisor on your dissertation topic (64%)*

2. *Supervisor's number of publications and/or presentations at scientific conferences (25%)*

3. *The alignment of the supervisor's research interests with yours (59%)*

4. *The supervisor has doctoral students who successfully defended their dissertations (32%)*

5. *Good feedback about the supervisor from other doctoral students or faculty (40%)*

6. *Successful previous experience of interaction with the supervisor (40%)*

7. *Friendship with the supervisor (25%)*

8. *There were no other faculty members who worked on your research topic (11%)*

The initial survey question also contained an open-ended "Other" option, which we excluded from the analysis because the category contained very few observations (N = 97, or 4% of the initial sample) and, more importantly, the responses were highly heterogeneous. To be used in the analysis, the presented options were merged into several components, which are referred to as selection strategies, using principal component analysis (PCA). We employed PCA on the tetrachoric correlation matrix since variables were presented as binary. Promax rotation was used because we expected different selection strategies to be correlated. Analysis was conducted in *R* using the *psych* package [51]. To determine the optimal number of components, we compared models with two to five components. Based on explained variance and interpretability, we selected the model with four components. The selected model explained 74% of the variance. The extracted components represent the following reasons used to select a supervisor: 1) subject-matter expertise, 2) successful supervision history, 3) positive personal relationships, and 4) the "only option." Factor loadings and communalities for the final model are presented in Table 1.

We then calculated component scores for each participant using the *Thurstone* method, which estimates scores by regressing observed variables on their loadings to maximize the correlation between the estimated scores and the true latent factors [51]. These component scores are standardized continuous variables and can be interpreted as to what extent each participant relied on various groups of criteria when selecting a supervisor. Higher scores indicate a stronger alignment with a particular group of criteria.

**Difficulties in interaction with a supervisor.** Our first dependent variable describes whether a student reports difficulties in interaction with a supervisor during their doctoral studies. To identify this, we used a general survey question

**Table 1. Factor loadings and communalities of the PCA model.**

| | RC1 | RC2 | RC3 | RC4 | Communalities |
|---|---|---|---|---|---|
| The authority and expertise of the supervisor on your dissertation topic | 0.701 | 0.072 | −0.152 | −0.247 | 0.58 |
| Supervisor's number of publications and/or presentations at scientific conferences | 0.688 | 0.351 | −0.035 | 0.246 | 0.66 |
| The alignment of the supervisor's research interests with yours | 0.825 | −0.265 | 0.171 | −0.003 | 0.78 |
| The supervisor has doctoral students who successfully defended their dissertations | −0.008 | 0.839 | −0.031 | −0.085 | 0.71 |
| Good feedback about the supervisor from other doctoral students or faculty | −0.064 | 0.855 | 0.114 | −0.020 | 0.75 |
| Successful previous experience of interaction with the supervisor | 0.065 | 0.000 | 0.820 | −0.129 | 0.69 |
| Friendship with the supervisor | −0.025 | 0.081 | 0.868 | 0.087 | 0.77 |
| There were no other faculty members who worked on your research topic | −0.015 | −0.065 | −0.023 | 0.950 | 0.91 |

*Note: RC1 stands for subject-matter expertise, RC2 is successful supervision history, RC3 is positive personal relationships, and RC4 corresponds to the "only option."*

("To what extent have the following factors hindered your progress in the doctoral program?"), which, among other options, included the "difficulties in interacting with the supervisor" option. In our sample, 22% of the participants reported difficulties in interacting with their supervisor (N = 1606, excluding missing data).

**A student's lack of confidence about defense.** As the second outcome, we used a standardized variable reflecting students' lack of confidence about completing their dissertation, which was proposed in previous research [43] and combined students' agreement with the following statements (measured on a scale from 1 to 4):

*"I am afraid…"*

1. *I will not defend my dissertation in time*

2. *I will not pass my next examination*

3. *All my work is being done in vain and does not get me closer to the defense*

4. *That my research does not correspond to the level of a doctoral dissertation*

5. *I am not capable of completing my dissertation*

Extraction of the component was carried out using categorical principal component analysis (CatPCA; in *R* using the *princals* class of the *Gifi* package [52]). The resulting model with one component explained 70% of the variance. Component loadings and communalities for this model are presented in Table 2. The resulting variable was a standardized continuous one with a higher value indicating students' lower confidence in the defense (N = 1497, excluding missing data).

**Control variables.** The first control variable indicates whether a student finds the frequency of meetings with a supervisor sufficient, which may be an indicator of matching working styles. For analytical purposes, we recoded this variable from ordinal into a binary format ("Fully sufficient" and "Rather sufficient" coded as 1; "Rather inadequate" and "Quite inadequate" coded as 0). Previous studies have highlighted the significance of this predictor and its impact on doctoral students' experiences [53]. The second control variable is the age of the supervisor, treated as an ordinal variable. This characteristic is directly related to the supervisor's scientific authority, supervisory experience, and position at the university. It may be associated with different styles of supervision and requirements expected of students. These two variables were used only in the models related to difficulties in interaction with a supervisor.

We considered students' employment status and, if employed, their type of employment during their doctoral training as another control variable. The vast majority of doctoral students in Russia (90%) are employed during their studies

**Table 2. Factor loadings and communalities for the CatPCA model.**

| I am afraid… | Loadings | Communalities |
|---|---|---|
| I will not defend my dissertation in time | 0.809 | 0.655 |
| I will not pass my next examination | 0.772 | 0.596 |
| All my work is being done in vain and does not get me closer to the defense | 0.872 | 0.760 |
| That my research does not correspond to the level of a doctoral dissertation | 0.838 | 0.702 |
| I am not capable of completing my dissertation | 0.883 | 0.780 |

[54]. The effects of employment on doctoral outcomes are heterogenous. For example, on-campus employment related to research duties during doctoral training can increase the likelihood of dissertation defense. In contrast, off-campus employment and non-research jobs at the university are negatively associated with completion of a dissertation [30,55]. In terms of satisfaction with supervision, employed doctoral students are usually considered nontraditional students (see more on the term: [56]), and such students are characterized by a more autonomous work style and higher levels of satisfaction with their overall doctoral experience [48]. Drawing on the design of previous studies [30,55], we categorized employment status into four types to control possible heterogeneous effects: unemployed (10%), employed outside the university (50%), employed in a research position within the university (14%), and employed in non-research positions within the university (26%). This variable was constructed from responses to several survey questions: "Are you currently employed?"; "Do you work at the same university where you are currently studying?"; and "What type of position do you hold?" Answer options included "research position," "teaching position," "administrative position," and "other."

Additionally, all of our models included formal control variables such as the year of study, form of study, form of financing, type of university (leading or not), field of study, and gender.

## Analysis

We propose two sets of models to examine the relationship between supervisor selection strategies and various outcomes. Each set contains several models into which we sequentially incorporated predictors and control variables as described in the previous subsection. We rely on this technique to ensure that the effects of our variables of interest are stable and do not disappear when considering other important factors.

The first set of models was constructed using binary logistic regression due to the nature of the dependent variable, which reflects difficulties in interaction with a supervisor. In these models, we also included control variables that may be related to satisfaction with the supervisor-supervisee interaction (see the "Control Variables" subsection).

The second set consists of linear regression models, as the dependent variable was the lack of doctoral students' confidence about their defense, which is represented by a standardized continuous variable.

Missing data in the models were handled using a listwise-deletion strategy separately for each model.

## Results

### Difficulties in interaction with supervisor

The first set of models predicts whether a student faces difficulties in interaction with their supervisor. Results of the corresponding logistic regression models are presented in Table 3 (odds ratios and standard errors). In the regression tables, we first present our variables of interest and then all the control variables (progressing from the substantial ones to the more formal ones).

Model 1 demonstrates that selecting supervisors based on their subject-matter expertise and having a pre-existing positive personal relationship both significantly reduce the odds of experiencing interaction difficulties during doctoral training. Specifically, a one standardized unit increase in the component related to the subject-matter expertise reduces the odds

**Table 3. Results of binary logistic regression (DV: facing difficulties in interaction with a supervisor, yes/no).**

| Variable | Model 1 | | Model 2 | | Model 3 | | Model 4 | |
|---|---|---|---|---|---|---|---|---|
| | Exp(B) | SE | Exp(B) | SE | Exp(B) | SE | Exp(B) | SE |
| Intercept | 0.165*** | 0.293 | 0.853 | 0.352 | 1.034 | 0.363 | 1.231 | 0.42 |
| Subject-matter expertise | 0.607*** | 0.061 | 0.669*** | 0.069 | 0.67*** | 0.069 | 0.657*** | 0.07 |
| Positive personal relationships | 0.649*** | 0.067 | 0.65*** | 0.074 | 0.664*** | 0.075 | 0.653*** | 0.076 |
| Successful supervision history | 0.84** | 0.064 | 0.857* | 0.071 | 0.838* | 0.072 | 0.839* | 0.072 |
| Only option | 1.625*** | 0.056 | 1.487*** | 0.064 | 1.512*** | 0.065 | 1.499*** | 0.065 |
| Satisfied with the frequency of meetings | | | 0.107*** | 0.156 | 0.104*** | 0.157 | 0.104*** | 0.158 |
| Age of supervisor (ref = "60 and more") | | | | | | | | |
| 39 and less | | | | | 0.636 | 0.263 | 0.616 | 0.264 |
| 40-49 | | | | | 0.68* | 0.191 | 0.673* | 0.192 |
| 50-59 | | | | | 0.75 | 0.186 | 0.751 | 0.187 |
| Employment status (ref = "not employed") | | | | | | | | |
| Employed in a research position | | | | | | | 1.158 | 0.289 |
| Employed in other position | | | | | | | 0.757 | 0.268 |
| Employed outside the university | | | | | | | 0.717 | 0.242 |
| 1st year | 0.812 | 0.227 | 0.687 | 0.251 | 0.703 | 0.252 | 0.748 | 0.255 |
| 2nd year | 0.757 | 0.228 | 0.704 | 0.253 | 0.696 | 0.253 | 0.734 | 0.255 |
| 3rd year | 0.984 | 0.236 | 0.863 | 0.263 | 0.863 | 0.263 | 0.88 | 0.265 |
| Full-time | 1.416 | 0.267 | 1.901* | 0.308 | 1.974* | 0.31 | 1.883* | 0.314 |
| Tuition-free | 1.157 | 0.231 | 1.108 | 0.258 | 1.104 | 0.259 | 1.105 | 0.262 |
| Female | 1.115 | 0.137 | 1.07 | 0.154 | 1.077 | 0.154 | 1.079 | 0.156 |
| Field of study (ref = "Mathematics and natural science") | | | | | | | | |
| Humanities | 1.035 | 0.213 | 0.871 | 0.241 | 0.877 | 0.242 | 0.941 | 0.247 |
| Education and pedagogy | 0.942 | 0.239 | 0.922 | 0.264 | 0.948 | 0.266 | 1.017 | 0.27 |
| Social science | 0.803 | 0.204 | 0.698 | 0.229 | 0.724 | 0.231 | 0.768 | 0.233 |
| Engineering and technology | 1.243 | 0.185 | 1.142 | 0.207 | 1.126 | 0.207 | 1.202 | 0.211 |
| Leading university | 1.283 | 0.143 | 1.169 | 0.16 | 1.176 | 0.161 | 1.137 | 0.162 |
| Observations | 1606 | | 1566 | | 1566 | | 1566 | |
| R^2 McFadden | 0.105 | | 0.220 | | 0.240 | | 0.244 | |

*Note: \*p < 0.05, \*\*p < 0.01, \*\*\*p < 0.001*

by 39%, while prioritizing positive personal relationships decreases the odds by 35% per unit. Additionally, having a supervisor with a successful supervision history also markedly decreases interaction difficulties. In contrast, selecting a supervisor out of necessity, when no other options are available, increases the odds of subsequent interaction difficulties by 63% per standardized unit.

In Models 2, 3, and 4, we included control variables concerning satisfaction with the frequency of meetings with a supervisor, the age of the supervisor, and the student's employment status. Despite the significant coefficients of some variables, the effects of the analyzed selection strategies remained significant, exhibiting similar effects as those observed in the first model.

## Lack of confidence about defense

The second set of models predicts students' level of uncertainty regarding the future defense. The results of the corresponding linear regression models are presented in Table 4.

Table 4. Results of linear regression analysis (DV: lack of confidence about future successful defense).

| Variable | Model 1 | | | | Model 2 | | | |
|---|---|---|---|---|---|---|---|---|
| | Estimate | SE | LB CI 95% | UB CI 95% | Estimate | SE | LB CI 95% | UB CI 95% |
| Intercept | −0.4*** | 0.1 | −0.6 | −0.19 | −0.32* | 0.14 | −0.58 | −0.05 |
| Subject-matter expertise | −0.18*** | 0.02 | −0.22 | −0.14 | −0.18*** | 0.02 | −0.22 | −0.13 |
| Positive personal relationships | −0.04 | 0.02 | −0.08 | 0.01 | −0.03 | 0.02 | −0.08 | 0.01 |
| Successful supervision history | −0.06** | 0.02 | −0.1 | −0.02 | −0.07** | 0.02 | −0.12 | −0.02 |
| Only option | 0.14*** | 0.02 | 0.1 | 0.19 | 0.15*** | 0.03 | 0.1 | 0.2 |
| 1st year | 0.01 | 0.09 | −0.16 | 0.17 | −0.09 | 0.09 | −0.27 | 0.09 |
| 2nd year | 0.13 | 0.09 | −0.04 | 0.3 | 0.08 | 0.09 | −0.1 | 0.26 |
| 3rd year | 0.12 | 0.09 | −0.05 | 0.3 | 0.12 | 0.1 | −0.07 | 0.31 |
| Employment status ref = "not employed" | | | | | | | | |
| Employed in a research position | | | | | −0.33** | 0.1 | −0.53 | −0.12 |
| Employed in other position | | | | | −0.04 | 0.09 | −0.22 | 0.15 |
| Employed outside the university | | | | | −0.05 | 0.09 | −0.22 | 0.12 |
| Full-time | 0.07 | 0.09 | −0.11 | 0.24 | 0.11 | 0.1 | −0.08 | 0.31 |
| Tuition-free | 0.05 | 0.08 | −0.11 | 0.2 | 0.08 | 0.09 | −0.09 | 0.25 |
| Female | 0.17*** | 0.05 | 0.07 | 0.26 | 0.17** | 0.05 | 0.06 | 0.27 |
| Field of study ref = "Mathematics and natural science" | | | | | | | | |
| Humanities | 0.18* | 0.08 | 0.03 | 0.33 | 0.11 | 0.09 | −0.06 | 0.27 |
| Education and pedagogy | 0 | 0.09 | −0.17 | 0.17 | −0.05 | 0.09 | −0.23 | 0.14 |
| Social science | 0.1 | 0.07 | −0.04 | 0.24 | 0.12 | 0.08 | −0.03 | 0.27 |
| Engineering and technology | 0.11 | 0.07 | −0.02 | 0.24 | 0.08 | 0.07 | −0.06 | 0.23 |
| Leading University | 0.16** | 0.05 | 0.05 | 0.26 | 0.19*** | 0.06 | 0.08 | 0.3 |
| Observations | 1497 | | | | 1497 | | | |
| Adj R² | 0.075 | | | | 0.082 | | | |

Note: *p < 0.05, **p < 0.01, ***p < 0.001

The first linear model demonstrates the significance of selection strategies based on subject-matter expertise, successful supervision history, and the "only option" factor. The first two factors are associated with a decrease in lack of confidence, with subject-matter expertise showing the strongest negative association; specifically, this strategy reduces the dependent variable by 0.18 per standardized unit, all other things being equal. Similarly, choice based on successful supervision history reduces the lack of confidence by 0.06 per standardized unit. However, selecting a supervisor as the "only option" is positively associated with a lack of confidence. Notably, reliance on positive personal relationships when choosing a supervisor does not significantly predict outcomes in this model, contrasting with previous logistic models predicting difficulties in supervisor interaction.

In the second model, we included employment status as a control variable. Working in a research position at their university is shown to decrease students' lack of confidence, consistent with findings from previous research [30,55]. The effects of the predictors related to supervisor selection strategies in Model 2 mirror those in the first model.

## Discussion

The purpose of the current study was to determine the relationship between various strategies doctoral students employ when selecting their supervisors and two key outcome variables: whether students face any difficulties in interaction with supervisors during their subsequent training and how confident they are in their future successful dissertation defense. Our findings contribute to the current literature on supervisor selection in several ways.

First, we demonstrated that selection based on the supervisor's academic expertise is the most effective among other possible strategies. By choosing supervisors based on their expertise in the dissertation topic, doctoral students reduce the likelihood of encountering problems with them during the training period and feel more confident regarding defending a dissertation. Additionally, positive personal relationships and successful supervision history have also shown significant associations in reducing the probability of facing difficulties with the supervisor. These results generally align with previous findings that emphasized the significance of academic attributes and work-oriented relationships [6,13,15,25]. In terms of practical implications, these findings imply that while "soft" factors such as personal compatibility may matter for day-to-day collaboration, institutional systems should prioritize transparent access to supervisors' academic records to empower evidence-based selection.

Second, we discovered that although selection based on positive personal relationships reduces the chances of facing difficulties with supervisors, we found no significant association between this component and students' levels of confidence about future defense. We interpret this finding as confirming the results of qualitative studies about bidirectional and complex effects of this criterion: positive relationships alone, without additional support such as strong academic expertise or successful supervision background, do not necessarily help students in navigating their doctoral challenges [18,27]. This suggests that while good interpersonal dynamics can create a more supportive environment, they may not compensate for deficiencies in substantive academic guidance when it comes to building confidence in one's research outcomes. The disassociation between relational comfort and defense confidence underscores the importance of distinguishing between socio-emotional support and research-specific and subject-specific mentoring in doctoral supervision.

Third, the only strategy that provides negative results regarding both facing difficulties with the supervisor and students' levels of confidence is the "only option" scenario (i.e., when students have no other choice but to work with a particular supervisor). As subject-matter expertise is controlled in all models, the negative coefficients are reached by virtue of the absence of alternatives, which aligns with the results of previous studies [18]. This scenario and the revealed negative effects can be interpreted in several ways. On the one hand, this scenario may reflect the general deficit of potential supervisors, which, considering the high requirements imposed on them [46] and the declining number of scientific degree holders in Russia, may indeed be relevant to some Russian universities. In this case, one potential practical solution may be introducing team or co-supervision practices, which have become standard in some countries [57] but are still scarce in Russia [43]. While one of the supervisors would have the necessary subject expertise and meet all the formal criteria, the second could be chosen from among younger scholars to provide methodological, organizational, or informational support, thus giving that scholar the experience necessary for future supervision. On the other hand, the "only option" scenario may also be explained by students' lack of familiarity with all relevant specialists within the university. In this case, universities and departments should provide more detailed information about the available supervisors and their topics of expertise, as well as their supervision history, and encourage students to make informed choices rather than constrained ones.

When interpreting the results, one needs to take into account the limitations and restrictions associated with this research.

First, we focus only on the criteria students use when selecting their final supervisor; we lack detailed information about the circumstances in which these selections are made. As mentioned in the national context section, it is not always the case that students themselves choose their supervisors; sometimes, supervisors are suggested and/or appointed by the department. We cannot differentiate between these alternatives based on our data. Thus, our findings are more relevant for national and institutional contexts in which students themselves choose their own supervisors.

Second, we do not know much about the nature, diversity, and extent of difficulties that students face when interacting with their supervisors (our first dependent variable). It is plausible that distinct selection strategies may be linked to diverse issues in interaction. Further investigation is required to delve deeper into these variations.

Third, as we rely on cross-sectional data, we do not make any conclusions about causal relationships between our variables, nor do we know how various selection strategies are related to the actual performance of doctoral students (because we do not have such variables).

Fourth, we could encounter a survival bias in our research as our survey does not cover students who, for any reason, dropped out of their program, particularly because of some kind of supervisor-supervisee mismatch or other issues related to supervisor interaction. This methodological constraint may lead to over- or under-estimation of some of the identified effects.

Finally, caution is warranted when generalizing our study's results, as the analysis is based on a non-probabilistic, quota-based national survey and contains inherent biases (e.g., the sample included a higher proportion of women, likely due to their greater willingness to participate). Our study does not aim to identify the dominant choice strategies in Russia; rather, it seeks to determine how these strategies may be associated with various outcomes.

## Conclusion

The literature on doctoral education has paid much attention to the supervisor-supervisee relationship, but relatively little to how that relationship begins. In addition, few studies have attempted to highlight supervisor selection strategies, and those that do offer limited analysis of the complex links between selection strategies and doctoral program outcomes. To address this gap, we explored the links between various doctoral outcomes and four possible strategies students use to select their supervisor, i.e., strategies based on (1) subject-matter expertise, (2) positive personal relationships, (3) successful supervision history, and (4) the "only option" scenario, which effectively signifies a lack of choice. Guided by our research questions and using regression analyses on the nationwide survey of doctoral students in Russia, we found a strong relationship between these selection strategies and two outcomes: the likelihood of encountering difficulties in interactions with a supervisor and a lack of confidence about the dissertation defense. As selection strategies, subject-matter expertise and successful supervision history are negatively related to both encountering difficulties with a supervisor and feeling uncertain about the defense. Basing selection on a positive personal relationship is negatively associated with difficulties in interactions with a supervisor, but it is not significantly linked to a lack of confidence. In contrast, the "only option" scenario is positively associated with both facing difficulties with supervisors and lacking confidence about the upcoming defense. The study highlights the critical role of agency in supervisor selection, showing that structured support for informed decision-making – whether through institutional matchmaking systems or student mentoring programs – can mitigate risks of poor supervision outcomes. These findings call for policy interventions that promote transparency in supervisor selection processes, especially in contexts where students have limited choices.

## Acknowledgments

The authors would like to thank Evgeniy Terentev, Nikita Smirnov, and Jennet Babaeva for their valuable comments on the initial version of the manuscript.

## Author contributions

**Conceptualization:** Svetlana Zhuchkova.

**Formal analysis:** Danila Pavliuk.

**Funding acquisition:** Svetlana Zhuchkova.

**Investigation:** Danila Pavliuk.

**Methodology:** Danila Pavliuk.

**Project administration:** Svetlana Zhuchkova.

**Supervision:** Svetlana Zhuchkova.

**Writing – original draft:** Danila Pavliuk, Svetlana Zhuchkova.

**Writing – review & editing:** Danila Pavliuk, Svetlana Zhuchkova.

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
