## [Decision Letter · Decision Letter 0]

PONE-D-24-33457
Choosing to succeed? Insights into doctoral students’ supervisor selection and its outcomes
PLOS ONE

Dear Dr. Pavliuk,

Thank you for submitting your manuscript to PLOS ONE. After careful consideration, we feel that it has merit but does not fully meet PLOS ONE’s publication criteria as it currently stands. Therefore, we invite you to submit a revised version of the manuscript that addresses the points raised during the review process.
 
Dear Authors, please carefully attend to the comments given by the reviewers and improve the quality of the manuscript. Its my earnest believe that the comments will enhance the quaity of the manuscript.

We look forward to receiving your revised manuscript.

Kind regards,

Bekalu Tadesse Moges

Academic Editor

PLOS ONE

Journal Requirements:

2.  We noted in your submission details that a portion of your manuscript may have been presented or published elsewhere. “We have published regression dataset that used in our research at Harvard Dataverse under CC0 v1. https://doi.org/10.7910/DVN/YKNXO3”

Reviewers' comments:

Reviewer's Responses to Questions

**Comments to the Author**

1. Is the manuscript technically sound, and do the data support the conclusions?

Reviewer #1: Yes

Reviewer #2: Yes

2. Has the statistical analysis been performed appropriately and rigorously? 

Reviewer #1: Yes

Reviewer #2: Yes

3. Have the authors made all data underlying the findings in their manuscript fully available?

Reviewer #1: No

Reviewer #2: Yes

4. Is the manuscript presented in an intelligible fashion and written in standard English?

Reviewer #1: No

Reviewer #2: Yes

5. Review Comments to the Author

Reviewer #1: Dear Authors

Thanks for the very interesting manuscript draft. I have read the draft and felt that - it is interesting and relevant to postgraduate supervision. If the suggested comments are effected, some of the academic institutions could benefit a lot from the manuscripts when considering guidelines for selecting the supervisor(s) by doctoral students.

1. Structure - please improve the structure by arranging your headings well. Presenting the section of data before materials and methods confuses the reader; combing the results and methods; and discussion with conclusion also confuses the reader.

2. Statistics - both the introduction and literature review could benefit a lot from additional statistics that relate to those who chose supervisors themselves, did they complete timeously or with a struggle, or those who were compelled to work with a particular supervisor, did they complete timeously, or did they have a good working relationship. What are the benefits of studying all these selection strategies? this is not clear in the introduction or background.

3. Tables - there are lots of tables that confuses the reader - consider compressing some of the tables. Additional, some of these tables are not referenced on the intext discussion.

4. Conclusion and recommendation are critical elements for a manuscripts - the current draft requires a separate clear conclusion and recommendations.

Reviewer #2: Thank you for the opportunity to review this manuscript about approaches to choosing a doctoral supervisor and relationships with supervisor challenges. I found the study interesting but quite hard to follow. I have a few overarching comments and several more comments within specific sections for the authors for consideration.

Overarching comments:

1. The manuscript needs a thorough review and edit for English language. For example, authors switch from plural to singular language but should stay plural.

2. Structure of sections is quite odd, - the distinction between the Methods and Results is unclear.

3. Methods section is not organized into traditional subsections such as Sample and Participants, Procedure, Measures, and Analysis.

4. Unclear if the study is about "Russian doctoral students" or "doctoral students in Russia", the latter which would include foreign born students in Russia.

Abstract

L11: "Research usually..." - which research?

Introduction

L36: Add citations to back up argument.

L40: "An inappropriate" - for clarity of meaning, probably better to say "unsuitable" or "unfitting".

L46: "..demographic factors" - give examples.

L50-51: "..various selection strategies contribute to the difficulties during doctoral training" - hard to follow argument, give examples.

L78: After "Doctoral students" add "usually" before "evaluate".

L85-87: This appears quite serious and would warrant more elaboration.

L89: "...supervisor's background... likely explained" - please add citations to back up argument.

L97: "These papers" - revise to "These studies".

L98: "..these factors can vary", Which factors and vary how? Clarify meaning.

L107: "..generalizing the obtained findings" - in Russia? Elsewhere? Your study is exclusive to Russia. How does that move the literature forward?

L115: "More than half of Russian doctoral students are male (66%)..". To be exact, this is 2/3.

L115: Note that the argument about 2/3 of Russian doctoral students being male and that the sample employed in this study is 53% female seriously calls to question the generalizability of the findings if unaddressed.

L117-119: Please give actual numbers. Statements about "challenges" in Russian doctoral education are unverified if left without actual trend numbers. For instance, what is the "crisis of Russian doctoral education"?

L123: "..supervisors in Russia are still the main figures in a doctoral journey.." - as opposed to what? Please clarify meaning.

L127-128: "..practices of team supervision" and "not regulated normatively" - please give examples for clarity.

L133-134: "..high academic requirements... 3 papers in highly ranked journals" - is this a requirement to defend a dissertation or some other milestone? Please clarify.

L136-139: Please explain context. Doctoral programs and procedures differ between places and countries. These are for example not the traditional processes in the US. Making sure that all audiences understand your narrative is important.

L145-146: Clarify meaning. Can you not become a supervisor if you are hired more than 30 days after each student's enrollment?

Data

This is where I would think the "Methods" section should begin.

L170: 2392 were in the numerator but the denominator is not included. Please clarify and insert a response rate.

L171: Why are students from "research institutes" excluded? Unsubstantiated to cut 15% of the sample in this way.

L183: 53% of the sample being female while 66% of doctoral students are male seriously calls to question the generalizability of the sample.

L184: with most participants being 1st or 2nd year, - how many of those have not yet selected a supervisor? Or if all already do than that should be clarified.

Materials and Methods

L188: Materials and Methods - this section reads like a "Measures" subsection.

L202: What about "other reasons" - was there no option for that?

L206: As far as I can tell, these are analytical results and should belong in the Results section.

L210-211: Unclear how PCA accounts for variations in response numbers. With multiple choice questions some students may have selected 1 option and some all options. Please clarify.

L223: Briefly expplain the Thurstone method.

L223-224: "These scores.." - which scores?

Table 3: Findings have unclear relevance to the study. Please clarify in the Introduction why these findings are important and part of the study.

Table 3: Last section in analysis. F-statistic assumes linear trends but the "Field of Study" is nominal variable. This analysis appears incorrect.

L252-253: "N=1606 excluding..". Unclear relevance and meaning.

L285: "A significant proportion" - unclear meaning. What is a "significant proportion".

Results

L304: By here a lot of "Results" have already been shown.

Tables 5 and 6: Why so many models and why this order of variables in the models? Please clarify.

6. PLOS authors have the option to publish the peer review history of their article (what does this mean?). If published, this will include your full peer review and any attached files.

Reviewer #1: **Yes: **Masenyani Oupa Mbombi

Reviewer #2: **Yes: **Alfgeir L. Kristjansson

---

## [Author Response · Author response to Decision Letter 1]

19 Apr 2025

We sincerely appreciate the valuable comments provided by the reviewers. We have carefully reviewed each comment, and we believe that their insightful feedback has greatly enhanced the revised version of our manuscript. Our detailed responses to each point can be found in the below and in the file "Response to Reviewers".

### Reviewer 1 ###

___

Reviewer’s Comment: Structure - please improve the structure by arranging your headings well. Presenting the section of data before materials and methods confuses the reader; combing the results and methods; and discussion with conclusion also confuses the reader.

Authors’ Response: We changed the structure of the manuscript. In particular, we merged the “Data” and “Materials and methods” sections into a single “Methods” section to create a more coherent framework for describing our procedures. Within this section, we re-labeled the “Data” as “Sample” for greater clarity in presenting our participants and data collection. We also separated the “Discussion” and “Conclusion” sections.

____

Reviewer’s Comment: Statistics - both the introduction and literature review could benefit a lot from additional statistics that relate to those who chose supervisors themselves, did they complete timeously or with a struggle, or those who were compelled to work with a particular supervisor, did they complete timeously, or did they have a good working relationship. What are the benefits of studying all these selection strategies? this is not clear in the introduction or background.

Authors’ Response: Unfortunately, after a thorough additional literature review we conclude that there are no available quantitative statistics on this matter. We provide more examples of qualitative studies of various consequences for the selection strategies and more explicitly emphasize the lack of quantitative studies, as well as the limitations of the existing ones.

____

Reviewer’s Comment: Tables - there are lots of tables that confuses the reader - consider compressing some of the tables. Additional, some of these tables are not referenced on the intext discussion.

Authors’ Response: We removed two tables from the manuscript that may have been misleading to readers. We also checked the remaining tables that we refer to in the text.

____

Reviewer’s Comment: Conclusion and recommendation are critical elements for a manuscripts - the current draft requires a separate clear conclusion and recommendations.

Authors’ Response: We separated the “Discussion” and “Conclusion” sections. Conclusion now only summarises our own results, while “Discussion” provides links with previous literature as well as practical recommendations based on our findings.

___

Reviewer’s Comment: Add the conclusion on the abstract

Authors’ Response: We added a conclusion to the abstract (“The study highlights the critical role of agency in supervisor selection, showing that structured support for informed decision-making – whether through institutional matchmaking systems or student mentoring programs – can mitigate risks of poor supervision outcomes. These findings call for policy interventions that promote transparency in supervisor selection processes, especially in contexts where students have limited choices.”).

___

Reviewer’s Comment: L48: what is the significance of studying the relationship between these factors

Authors’ Response: We added the explanation: “Understanding this linkage is crucial because early-stage selection decisions may either facilitate smooth progress or lead to irreversible conflicts, ultimately impacting completion rates and research quality.”

___

Reviewer’s Comment: L58: Add a background relating to the benefits of selecting the supervisors, either by the institutional or personal factors

Authors’ Response: For each of the reviewed criteria to choose a supervisor, we added explanations on how a student can benefit from their interaction.

___

Reviewer’s Comment: L98: This is a good sentence, consider incorporating the same to the above paragraphs

Authors’ Response: Thanks for this suggestion.

___

Reviewer’s Comment: L117: Any statistics about those selecting supervisors for themselves or those who supervisors were chosen for them.

It can be interesting if you can find the existing statistics that shows differences in terms of completion or study progress between those who chose supervisors themselves and those whom the supervisors were chosen for them.

Authors’ Response: Unfortunately, after a thorough additional literature review we conclude that there are no available quantitative statistics on this matter. We provide more examples of qualitative studies of various consequences for the selection strategies and more explicitly emphasize the lack of quantitative studies, as well as the limitations of the existing ones. Likewise, we cannot provide up-to-date and representative statistics on which is more common in Russia: students choosing their supervisors or their being appointed by the department. However, we can indirectly assert that doctoral students more frequently select their own supervisors, as large-scale surveys indicate that only 16% choose them based on departmental recommendations. We have clarified this point in the text.

___

Reviewer’s Comment: L155: This arrangement of data before methods is confusing to me as reader - consider incorporating this section to the method section

Authors’ Response: We changed the structure of the manuscript. In particular, we merged the “Data” and “Materials and methods” sections into a single “Methods” section to create a more coherent framework for describing our procedures. Within this section, we re-labeled the “Data” as “Sample” for greater clarity in presenting our participants and data collection.

___

Reviewer’s Comment: L156: Please unpack the survey - how was it like? Has the survey been tested for reliability and validity? What type of questions were included in the survey.

Authors’ Response: We significantly extended our description of the survey.

___

Reviewer’s Comment: L243: Where are you referring or narrating this table

Authors’ Response: We decided to remove this table and this part of our paper as non-relevant for our research.

___

Reviewer’s Comment: L345: What conclusion can you draw based on your findings? what advice/s can you give to institutions regarding doctoral supervisor selection? Can institution change their current practice?

Separate conclusion from discussion. make recommendations based on your findings.

The article structure require major adjustments. I struggle to follow when reading the article.

Authors’ Response: We separated the “Discussion” and “Conclusion” sections. Conclusion now only summarises our own results, while “Discussion” provides links with previous literature as well as practical recommendations based on our findings. We also extended this part of the article.

___

Reviewer’s Comment: L348: what is your conclusion based on this statement?

Authors’ Response: We included main conclusions after each finding.

___

Reviewer’s Comment: L412: Which reference style is practiced here? Be consistent as well

Authors’ Response: We checked the References section for consistency with the style of the journal and corrected inaccuracies.

### Reviewer 2 ###

___

Reviewer’s Comment: The manuscript needs a thorough review and edit for English language. For example, authors switch from plural to singular language but should stay plural.

Authors’ Response: We relied on formal proofreading procedures in the new version of the manuscript.

_____

Reviewer’s Comment: Structure of sections is quite odd, - the distinction between the Methods and Results is unclear.

Authors’ Response: We significantly restructured our methodological section of the manuscript, as well as removed irrelevant descriptive results from this section.

_____

Reviewer’s Comment: Methods section is not organized into traditional subsections such as Sample and Participants, Procedure, Measures, and Analysis.

Authors’ Response: We changed the structure of the manuscript. In particular, we merged the Data and Materials and methods sections into a single Methods section to create a more coherent framework for describing our procedures. This section is now divided into more traditional Sample, Measures, Analysis subsections.

_____

Reviewer’s Comment: Unclear if the study is about "Russian doctoral students" or "doctoral students in Russia", the latter which would include foreign born students in Russia.

Authors’ Response: We refined the text for clarity. Our study encompasses doctoral students educated in Russia, including those who were born outside the country.

_____

Reviewer’s Comment: L11: "Research usually..." - which research?

Authors’ Response: We clarified in the manuscript that the discussion pertains to the field of doctoral supervision.

_____

Reviewer’s Comment: L36: Add citations to back up argument.

Authors’ Response: We added citations to support this argument.

_____

Reviewer’s Comment: L40: "An inappropriate" - for clarity of meaning, probably better to say "unsuitable" or "unfitting".

Authors’ Response: We made the suggested replacement.

_____

Reviewer’s Comment: L46: "..demographic factors" - give examples.

Authors’ Response: We included examples of demographic factors in the Introduction.

_____

Reviewer’s Comment: L50-51: "..various selection strategies contribute to the difficulties during doctoral training" - hard to follow argument, give examples.

Authors’ Response: We clarified in the text that this refers to difficulties in interacting with the supervisor.

_____

Reviewer’s Comment: L78: After "Doctoral students" add "usually" before "evaluate".

Authors’ Response: We added the suggested word.

_____

Reviewer’s Comment: L85-87: This appears quite serious and would warrant more elaboration.

Authors’ Response: We added more discussion and references on this matter.

_____

Reviewer’s Comment: L89: "...supervisor's background... likely explained" - please add citations to back up argument.

Authors’ Response: We added citations to support this argument.

_____

Reviewer’s Comment: L97: "These papers" - revise to "These studies".

Authors’ Response: We made the suggested replacement.

_____

Reviewer’s Comment: L98: "..these factors can vary", Which factors and vary how? Clarify meaning.

Authors’ Response: We revised this part of the paragraph to clarify its meaning.

_____

Reviewer’s Comment: L107: "..generalizing the obtained findings" - in Russia? Elsewhere? Your study is exclusive to Russia. How does that move the literature forward?

Authors’ Response: We clarified some features of our study that may improve literature in this part of text. In particular, we shifted our focus from generalization to the fact that we present a larger sample from a nationwide survey that covers different categories of doctoral students and different organizations and allows us to conduct quantitative analysis of several factors simultaneously. We additionally added how our study could be useful to policy-makers.

_____

Reviewer’s Comment: L115: "More than half of Russian doctoral students are male (66%)..". To be exact, this is 2/3.

Authors’ Response: We don’t use these exact statistics in the updated version of the paper (see next comment for details).

_____

Reviewer’s Comment: L115: Note that the argument about 2/3 of Russian doctoral students being male and that the sample employed in this study is 53% female seriously calls to question the generalizability of the findings if unaddressed.

Authors’ Response: In our original draft, we presented the last and actual figures for the 2022-2023 study year. However, because our survey sample was collected at the end of the 2021-2022 academic year (May-August 2022), when the proportion of male doctoral students across all types of organisations was 56%, we recognized the potential for confusion. To maintain consistency in our statistical reporting, we therefore used data from the 2021–2022 academic year and substantiated these figures by citing an authoritative statistical compendium. Additionally, we extended our description of limitations to warn readers on potential biases produced by overrepresentation of females.

_____

Reviewer’s Comment: L117-119: Please give actual numbers. Statements about "challenges" in Russian doctoral education are unverified if left without actual trend numbers. For instance, what is the "crisis of Russian doctoral education"?

Authors’ Response: We provided relevant statistical data to this paragraph.

_____

Reviewer’s Comment: L123: "..supervisors in Russia are still the main figures in a doctoral journey.." - as opposed to what? Please clarify meaning.

Authors’ Response: We clarified this part (“supervisors in Russia are still the main figures in a doctoral journey, i.e., they still provide all the guidelines on how to prepare and defend dissertation as opposed to the structured models where some of the necessary knowledge could be provided by doctoral school or other faculty”).

_____

Reviewer’s Comment: L127-128: "..practices of team supervision" and "not regulated normatively" - please give examples for clarity.

Authors’ Response: We clarified this part (“practices of team supervision (i.e., cases when a student has several official supervisors) are rare and can only be possible in circumstances of multidisciplinary research or on programs arranged by two institutions”).

_____

Reviewer’s Comment: L133-134: "..high academic requirements... 3 papers in highly ranked journals" - is this a requirement to defend a dissertation or some other milestone? Please clarify.

Authors’ Response: We clarified this part (“to get to a dissertation defence”).

_____

Reviewer’s Comment: L136-139: Please explain context. Doctoral programs and procedures differ between places and countries. These are for example not the traditional processes in the US. Making sure that all audiences understand your narrative is important.

Authors’ Response: We tried to clarify the specifics of the Russian context in this part: “Thus, defences in Russia follow strict formal requirements mandated by the Higher Attestation Commission, including mandatory pre-defence procedures (which require departmental reviews, approval timelines, formal announcements on the websites), document verification at multiple administrative levels, standardized formatting rules for dissertation manuscripts, etc. Usually it is supervisors who navigate these bureaucratic flows, i.e., interpret these requirements for candidates, coordinate with dissertation councils, as well as secure necessary approvals.”

_____

Reviewer’s Comment: L145-146: Clarify meaning. Can you not become a supervisor if you are hired more than 30 days after each student's enrollment?

Authors’ Response: We clarified this part (“it is obligatory for each student to choose and approve a supervisor within one month after the start of the program”).

_____

Reviewer’s Comment: This is where I would think the "Methods" section should begin.

Authors’ Response: We merged the Data and Materials and methods sections into a single Methods section to create a more coherent framework for describing our procedures. Within this section, we re-labeled the Data subsection as Sample for greater clarity in presenting our participants and data collection.

_____

Reviewer’s Comment: L170: 2392 were in the numerator but the denominator is not included. Please clarify and insert a response rate.

Authors’ Response: The survey employed a quota‐based sampling approach (this information, along with the established quotas, is detailed in the “Sample” subsection). Invitations were distributed in a manner that prevented determining the exact number of individuals who received or viewed the invitation link (university administrators were responsible for that, not the authors of the paper). Because a precise denominator is unavailable, a response rate cannot be computed and is not applicable to our data.

To enhance the transparency of our research and the characteristics of our data, we have incorporated an improved description of the survey data and collection methods

---

## [Decision Letter · Decision Letter 1]

Choosing to succeed? Insights into doctoral students’ supervisor selection and its outcomes

PONE-D-24-33457R1

Dear Dr. Danila Pavliuk,

We’re pleased to inform you that your manuscript has been judged scientifically suitable for publication and will be formally accepted for publication once it meets all outstanding technical requirements.

Kind regards,

Bekalu Tadesse Moges

Academic Editor

PLOS ONE

Additional Editor Comments (optional):

Reviewers' comments:

Reviewer's Responses to Questions

**Comments to the Author**

1. If the authors have adequately addressed your comments raised in a previous round of review and you feel that this manuscript is now acceptable for publication, you may indicate that here to bypass the “Comments to the Author” section, enter your conflict of interest statement in the “Confidential to Editor” section, and submit your "Accept" recommendation.

Reviewer #2: All comments have been addressed

2. Is the manuscript technically sound, and do the data support the conclusions?

Reviewer #2: Yes

3. Has the statistical analysis been performed appropriately and rigorously? 

Reviewer #2: Yes

4. Have the authors made all data underlying the findings in their manuscript fully available?

Reviewer #2: Yes

5. Is the manuscript presented in an intelligible fashion and written in standard English?

Reviewer #2: Yes

6. Review Comments to the Author

Reviewer #2: The authors have addressed my earlier comments to my satisfaction. I commend them on a job well done. Thank you

7. PLOS authors have the option to publish the peer review history of their article (what does this mean?). If published, this will include your full peer review and any attached files.

Reviewer #2: No

---

## [Editor Report · Acceptance letter]

PONE-D-24-33457R1

PLOS ONE

Dear Dr. Pavliuk,

I'm pleased to inform you that your manuscript has been deemed suitable for publication in PLOS ONE. Congratulations! Your manuscript is now being handed over to our production team.

Kind regards,

on behalf of

Dr. Bekalu Tadesse Moges

Academic Editor

PLOS ONE